# Exposure of CuO Nanoparticles Contributes to Cellular Apoptosis, Redox Stress, and Alzheimer’s Aβ Amyloidosis

**DOI:** 10.3390/ijerph17031005

**Published:** 2020-02-05

**Authors:** Ying Shi, Alexander R. Pilozzi, Xudong Huang

**Affiliations:** Neurochemistry Laboratory, Department of Psychiatry, Massachusetts General Hospital and Harvard Medical School, Charlestown, MA 02129, USA; yshi@rics.bwh.harvard.edu (Y.S.); apilozzi@mgh.harvard.edu (A.R.P.)

**Keywords:** nanoparticles, engineered nanomaterials, Alzheimer’s disease, apoptosis, reactive oxygen species, Aβ

## Abstract

Fe_2_O_3_, CuO and ZnO nanoparticles (NP) have found various industrial and biomedical applications. However, there are growing concerns among the general public and regulators about their potential environmental and health impacts as their physio-chemical interaction with biological systems and toxic responses of the latter are complex and not well understood. Herein we first reported that human SH-SY5Y and H4 cells and rat PC12 cell lines displayed concentration-dependent neurotoxic responses to insults of CuO nanoparticles (CuONP), but not to Fe_2_O_3_ nanoparticles (Fe_2_O_3_NP) or ZnO nanoparticles (ZnONP). This study provides evidence that CuONP induces neuronal cell apoptosis, discerns a likely p53-dependent apoptosis pathway and builds out the relationship between nanoparticles and Alzheimer’s disease (AD) through the involvement of reactive oxygen species (ROS) and increased Aβ levels in SH-SY5Y and H4 cells. Our results implicate that exposure to CuONP may be an environmental risk factor for AD. For public health concerns, regulation for environmental or occupational exposure of CuONP are thus warranted given AD has already become a pandemic.

## 1. Introduction

Nanotechnology is a rapidly developing field which involves the creation and/or manipulation of materials at the nanometer (nm) scale and is arising as a consequence of the novel properties these materials have to offer. According to the ASTM (formerly known as the American Society for Testing and Materials) standard definition, nanoparticles (NP) are particles with lengths that range from 1 to 100 nanometers in two or three dimensions [1]. Newly developed nanoparticle-based products have been introduced or are being currently developed in a number of diverse areas, including electronics, food, clothing, medicines, cosmetics and sporting equipment [2]. In medical fields, the applications include drug-delivery platforms [3,4], enhanced image contrast agents [5,6], chip-based nanolabs capable of monitoring [5] and manipulating individual cells [7], and probes that track the movements of cells [8] and individual molecules [9] as they move about in their environment. Some of the more common engineered nanomaterials (ENMs) used are comprised of metals and metal oxides of Al, Fe, Au, Si, Pd, Ce, Cu, Zn and Ti. Metal oxides, such as CuO, ZnO and Fe_2_O_3_ play a very important role in many areas of chemistry, physics, materials science and even biomedical fields.

Because of the wide application of nanoparticles and their highly publicized advantages in industrial and medical applications, human exposure to engineered nanoparticles, via inhalation, oral, dermal and injection routes, has increased. Inhalation of nanoparticles may occur as a consequence of their release into the environment, either during their manufacture or utilization. There is also an opportunity for their inhalation by workers during their manufacture [2,10]. Inhaled nanoparticles have the capacity to transit to the brain, routing along the olfactory nerve [11]. Although the properties of metal oxide nanoparticles may be impressive from a material science perspective, they may have overlooked toxic effects. Several studies have compared the toxicity of nanoparticles, some with a focus on metals or metal oxides [12]. Copper oxides in particular, from which cellular copper ions Cu^2+^ can be derived, are notable for their role in oxidation reactions, where they are reduced to Cu^1+^ [13]. CuONP cytotoxicity has been proven in many experiments on a cellular level [14,15], and in microbes [16,17,18]. They have been shown to be genotoxic, inducing DNA fragmentation and chromosomal damage on neuronal cell populations [19]. ZnONP toxicity has also been studied in cells and bacteria [20,21,22]. Published studies showed that ZnONPs were toxic to T-cells above 5 mM concentration [23] and toxic to neuroblastoma cells above 1.2 mM [12]. Thus, it appears that toxicity to the neuro cells were greater than toward bacteria, which have a threshold of 5 mM [22], while the opposite was true for T-cells. A study by Colon et al. showed that ZnONP actually improved normal osteoblast function, indicating non-toxicity [23]. Clearly the type of cell in question is important when considering the human cell toxicity of ZnONP [22]. Fe_2_O_3_/Fe_3_O_4_NP are used in medicine as both MRI contrast agents and anticancer agents [24]; they have been shown to exhibit limited toxicity [14]. However, some studies indicate they contribute to neurodegenerative processes, such as oxidative stress and apoptosis, following prolonged exposure [25,26]. Metal nanoparticles have also been found to be associated with the accumulation of Aβ and promoting the formation of amyloid plaques [27,28].

Nanoparticles of metal oxides produce reactive oxygen species (ROS), causing oxidative stress and ultimately leading to toxicological injury. Once inside the cell, nanomaterials may induce intracellular oxidative stress by disturbing the balance between oxidant and antioxidant processes. The oxidative stress induced by exposure to nanomaterials may stimulate an increase of the cytosolic calcium concentration [29] and/or it may cause the translocation of transcription factors (e.g., NF-κB) to the nucleus, which regulate pro-inflammatory genes, such as TNF-α and iNOS [30]. It has been reported that TiO_2_NP can decrease the level of glutathione, an antioxidant, in rat alveolar macrophages [31]. The ultrafine particles may exert mediated proinflammatory effects and induce the cell’s apoptosis through an ROS-mediated mechanism [32].

Alzheimer’s disease (AD) is a neurodegenerative disease that affects one in 10 persons over the age of 65 and 32% of people over the age of 85 [33]. It is associated with extracellular deposition of amyloid β-protein (Aβ) in the form of amyloid senile plaques, intracellular accumulation of neurofibrillary tangles and progressive neuronal loss [34]. Many of the neurons that undergo apoptosis in AD exhibit high levels of activated apoptotic proteins such as caspase 3 and Bax, and they exhibit neurofibrillary tangle pathology [35,36]. In addition, DNA damage and upregulation of the proapoptotic proteins p53 and Bax occur in vulnerable neuronal populations at a relatively early stage in the disease process [34]. Apoptosis may be responsible for cell disappearance and may play a key role at the root of the AD process [37]. Besides the pathological hallmarks of Aβ plaques and neurofibrillary tangles, AD brains exhibit evidence of ROS-mediated injury [38]. ROS, also called free radicals, are formed under normal conditions. An imbalance between the generation of free radicals and antioxidants may be involved in the pathogenesis of most neurodegenerative diseases [34], and ROS can contribute to the induction of apoptosis [39].

There is a great deal of research tying nanoparticles specifically to neurotoxic effects, especially with regard to CuONP. CuONP have been found to trigger significant cell-death events in human neuroglioma cells at concentrations of over 100 μm [15]. Studies indicate that CuONP exert a similar dose-dependent toxicity on H4 cells [15]. As of now, the cause of AD is not very clear. A host of different genetic and environmental factors appear to play into AD development. Some toxic nanoparticles that induce ROS generation, and lead to neuro-cell apoptosis, may be one of the risk factors for AD. In this study, we seek to assess the cytotoxicity of CuONP, ZnONP and Fe_2_O_3_NP, and determine whether CuONPs can induce brain-cell death via an apoptosis pathway and/or if they increase Aβ and/or ROS. In this study, we made use of the H4 human neuroglioma cell line, which is known to be impacted by CuONP [15]. We also utilized rat pheochromocytoma-12 cells (PC12), another popular neuron-like cell line [40], and SH-SY5Y neuroblastoma cells, which have wide use in the study of neurodegenerative disorders such as AD and Parkinson’s disease (PD) [41]. Herein, we demonstrated cellular exposure of CuONP may be implicated in AD amyloid pathology via neuronal apoptosis, ROS generation and increases in Aβ levels.

## 2. Materials and Methods

### 2.1. Cells Lines and Treatment

The human SH-SY5Y neuroblastoma cells (CRL-2266), human neuroglioma H4 (HTB-148) and rat PC12 (CRL-1721) were obtained from the American Type Culture Collection (ATCC, Manassas, VA, USA). They were grown in Dulbecco’s Modified Eagle Medium (DMEM) supplemented with 10% fetal bovine serum (FBS), 2 mM L-glutamine and 1% penicillin and streptomycin (all from Invitrogen, Carlsbad, CA, USA) at 37 °C in a humidified atmosphere containing 5% carbon dioxide. The cells were routinely passaged every 5 to 7 days. Nanoparticles were obtained from Sigma-Aldrich (St. Louis, MO, USA), and all were kept as a stock solution in 0.9% saline.

### 2.2. Cell Cytotoxicity Assay

Cytotoxicity tests were performed using the Promega CellTiter 96 Aqueous Non-Radioactive Cell Proliferation (MTS) assay to determine the number of viable cells in culture (Promega, Madison, WI, USA). Monolayers of 5000 cells were plated into 96-well flat-bottom cell culture plates (Corning Incorporated, Corning, NY, USA) in medium containing 10% FBS. Plating 24 h later, when the cells had attached and reached above 50% confluency, the medium was placed in a solution containing different nanoparticles at varying concentrations. All experiments were performed for 48 h and in triplicate. Controls included 0.1% saline. Nanoparticles were used at concentrations ranging from 0.01–100 μM. Each experiment was reproduced and confirmed in at least three independent experiments. After treatment, one solution reagent 20 μL/well was added directly to culture wells. After 3 h incubation, the plate was read by Benchmark Plus microplate spectrophotometer (Bio-Rad, Hercules, CA, USA). The absorbance of the formazan product was at 490 nm. The data was plotted and analyzed using Microsoft Excel (Redmond, WA, USA).

### 2.3. Trypan Blue Staining

H4 cells 1.5 × 10^5^, SH-SY5Y 2.5 × 10^5^ were plated into a 48-well plate. After treatment by different nanoparticles of different concentrations and at different times, cells were harvested with 10% medium and 1:2 dilution with 0.4% Trypan Blue solution (Sigma-Aldrich, St. Louis, MO, USA). Dead cells were stained blue, and the total living cells were counted on a hemocytometer under a microscope. Cells were adjusted in 200–500/squares with different dilutions. Then living cells were compared with cells without treatment.

### 2.4. TUNEL Staining

The TdT-mediated dUTP nick-end labeling (TUNEL) assay (TMR-RED in situ cell death detection kit) (Roche Applied Science, Penzberg, Germany) was carried out as per the manufacturer’s instructions. H4 cells: 2.0 × 10^4^, SH-SY5Y cells: 4.0 × 10^4^ and PC12 cells 4.0 × 10^4^ were plated into an 8-well glass chamber slide System (Fisher Scientific, Hampton, NH, USA) and grown for 24 h. Then cells were treated with CuONP 100 μM or medium (CTL). All experiments were performed for 24 h. After treatment, cells were washed with cold PBS and fixed with 4% paraformaldehyde 1 h, then incubated with permeabilization solution for 5 min on ice; cells were incubated with enzyme mixture for one hour in the humidified atmosphere for 60 min at 37 °C in the dark. Then Hoechst (Invitrogen, Carlsbad, CA, USA) was used to stain total nuclear. The optical sections were taken with a fluorescent microscope and the number of TUNEL-positive cells were counted.

### 2.5. Western Blots

SH-SY5Y cells 1.0 × 10^6^, H4 cells 8 × 10^5^ were plated and grown on 60 mm culture plate dishes for 24 h. When the cells had attached and reached above 60%–80% confluency, cells were treated with 100 µM CuONP at different times (SH-SY5Y 16 h and H4 6 h). After washing with cold PBS, cells were subjected to lysis in a RIPA buffer 0.1 mL (Boston BioProducts, Ashland, MA, USA) and SH-SY5Y 0.2 mL was applied to H4 cells. Each amount of protein lysate was loaded onto 4%–12% Tris-Bis gel (Invitrogen, Carlsbad, CA, USA) and then electro-transferred to a nitrocellulose membrane. The membranes were blocked with phosphate-buffered saline (PBS) containing 5% nonfat dry milk and 0.1% Tween 20 for 1 h at room temperature and incubated with 1:200 different primary antibodies (p53, 1:1000) overnight at 4°C. After washing with PBST, the membranes were incubated with secondary antibodies 1:5000 for 2 h at room temperature and developed a film by detection kit (Pierce Biotechnology, Rockford, IL, USA). Each experiment was repeated independently. Primary antibodies for p53, procaspase 3 and procaspase 9 were purchased from Santa Cruz Biotechnology (Dallas, TX, USA). Then the membranes were stripped with stripping buffer and re-incubated with the primary antibody for actin as a protein loading control. The final films were scanned, and densities were analyzed by NIH ImageJ software.

### 2.6. Caspase 3 Activity Assay

After treatment with CuONP (100 μM), 2.0 × 10^6^ cells (SH-SY5Y, PC12 and H4) were used to measure caspase 3 activity. After washing with ice PBS, cells were lysed by Assay Buffer (AB) 100 μL containing 50 mM 4-(2-hydroxyethyl)-1-piperazineethanesulfonic acid (HEPES) buffer, 100 mM NaCl, 1 mM EDT, 0.1% CHAPS, 1% glycerol, 10 mM DTT, and put on ice for 1 min, and then centrifuged at the highest speed for 1 min at room temperature. The supernatants were saved in −20 °C. Caspase 3 activity was determined using a caspase 3 activity detection kit (Millipore, Burlington, MA, USA). The caspase 3 fluorometric substrate (Ac-Asp-Glu-Val-Asp-AMC) was prepared as a stock solution of 7.2 mM in DMSO, stored in −20 °C, and used at a final concentration of 192 µM after dilution in AB. At the start of the proteolytic assay, cell lysate 50 µL and AB-diluted substrate 50 µL were added to a 96-well plate (Nunc), and fluorescence of the cleavage product was repeatedly measured over time at room temperature for 1 h and 30 min in a Wallac 1420 Multilabel Counter plate reader (PerkinElmer, Waltham, MA, USA), with the excitation wavelength at 350 nm and the detection wavelength at 460 nm. Maximal enzyme activity (*V*_max_) was calculated and expressed as relative fluorescence units/second. Data was plotted and analyzed using Microsoft Excel (Redmond, WA, USA).

### 2.7. Cellular Thiol Level Assay

The Fluorescent Thiol detection assay was performed as per manufacturer’s instructions. After treatment with CuONP (0.1–100 μM) and L-Buthionine-sulfoximine (L-BSO) (0.001–1 μM), 4.0 × 10^5^ cells (SH-SY5Y) and 4.0 × 10^5^ (PC12 and H4) were used to measure caspase 3 activity. Cells were washed by 2 mL PBS twice. Cellular thiol level was measured using the Fluorescent Thiol Detection Kit (Cell Technology Inc, Fremont, CA, USA). Cell lysis buffer 200 μL was added, and cell lysates were centrifuged at the highest speed for 5 min. The supernatants 50 μL and the reaction cocktail 50 μL (1 mL cell lysis buffer and 20 μL of the reconstituted dye) were added together into a 96-well black plate. The plate was read immediately with excitation at 485 nm and emission at 538 nm. The percentage of fluorescence unit change is calculated by each value divided by the control level. Data was plotted and analyzed using Microsoft Excel (Redmond, WA, USA).

### 2.8. Amyloid Aβ Level Measurement

Three cell lines, 1.0 × 10^6^ cells (SH-SY5Y) and 4.0 × 10^5^ (PC12 and H4), were plated on the Capture Antibody Coated Plate, and media 1.5 mL was added into the cells after plating 24 h. H4 and PC12 cells were treated for 6 h and SH-SY5Y cells were treated for 16 h. Media were taken out and saved in −80 °C. Aβ40 and Aβ42 levels were measured by Beta Amyloid x-40 and x-42 ELISA kit (Covance Research Products Inc., Princeton, NJ, USA). Samples were diluted 1 time and with a horseradish peroxidase (HRP) detection antibody: x-40, 1:1000; x-42, 1:250, overnight. After washing 5 times, mix chemiluminescent substrates 100 μL were added. The plate was read using the Luminescence mode on the Molecular Devices SpectraMax M5^e^ instrument. The percentage of luminescence unit change was calculated by each value divided by the control. Data was plotted and analyzed using Microsoft Excel (Redmond, WA, USA).

## 3. Results

### 3.1. Nanoparticles Inhibited Cell Proliferation

To determine the toxicity of the nanoparticles, we used a 3-(4,5-dimethylthiazol-2-yl)-5-(3-carboxymethoxyphenyl)-2-(4-sulfophenyl)-2H-tetrazolium (MTS) assay. Figure 1 shows the results of cell proliferation after treating three neuro cell lines, SH-SY5Y, H4 and PC12, with three different nanoparticles. The concentrations of the three NPs of CuONP, ZnONP and Fe_2_O_3_NP ranged from 0.01 to 100 μM. We observed that only high-dose CuONP has an anti-proliferative effect, and almost 90% inhibition on H4 and PC12 cells was observed after CuONP treatment of 48 h. We didn’t observe any significant difference between control and treatment of ZnONP and Fe_2_O_3_NP_._ Comparing how the three cell lines were affected by CuONP, H4 appears to be the most sensitive and SH-SY5Y appears to be the most tolerant of CuONP treatment. ZnONP and Fe_2_O_3_NP did not display cytotoxic effects. 

### 3.2. Nanoparticle Induced Neuro Cell Death

To test the effect of nanoparticles on cells, we used Trypan Blue dye staining to identify living cells. Figure 2 shows the results of the percentage of living cells after treating the three neuro cell lines with three different nanoparticles. The CuONP treated cells were observed at 2, 4, 6, 16 and 24 h and with concentrations ranging from 0.01 to 100 μM. We found the same result as the MTS assay: only high doses of CuONP induced cell death, but neither ZnONP nor Fe_2_O_3_ caused cell death at the concentrations tested (Figure 2A–C). Comparing the different effects on the three cell lines (Figure 2D) showed that H4 and PC12 cells started to die from 2 h and almost all cells had expired by 24 h, but in SH-SY5Y cells viability was reduced during the first 6 h of treatment, with 60% still alive 24 h after treatment. CuONP exhibited the most toxicity on H4 and PC12 cell lines while SH-SY5Y cells were the most resistant to CuONP toxicity.

### 3.3. CuONP Induced Cell Apoptosis

Based on the finding above, we tried to determine whether these cells underwent cell apoptosis due to treatment with CuONP. TUNEL assay detects the fragmentation of DNA which is characteristic of cells undergoing apoptotic cell death. As shown in Figure 3, the percentage of TUNEL-positive cells significantly increased. After treatment with CuONP, 20% of SH-SY5Y and almost 60% of H4 and PC12 of cells displayed TUNEL-positive staining, whereas only less than 1% of the control cells were TUNEL-positive. It showed that CuONP induced cell apoptosis on three cell lines after 24 h treatment at 100 μM. Copper ions are notably biologically important and powerful oxidizers [13], with CuONPs notably inducing a profound effect on ROS generation [42]. Perhaps it is this property that underlies the vast difference in toxicity between them and the other particles tested in this study.

### 3.4. CuONP Increased Caspase 3 Activity

To investigate the pathway of cell apoptosis by CuONP, we measured caspase 3 activity as an indicator of apoptosis induction since different upstream pathways leading to apoptosis depend on caspase 3 induction for final apoptotic execution. Figure 4 shows the effect of CuONP at the level of caspase 3 induction. We observed that CuONP induced caspase 3 activity levels 150%, 210%, and 355% of controls in SH-SY5Y, H4 and PC12 cell lines, respectively. This shows that after treatment of CuONP, caspase 3 activity significantly increased.

### 3.5. CuONP Decreased Procaspase 3, Procaspase 9 Expression

To further confirm caspase 3’s involvement in the process of apoptosis, we measured the changes in procaspase 3 and procaspase 9 following treatment with CuONP. Figure 5 shows that CuONP downregulated procaspase 3, and Figure 6 shows procaspase 9 expression after CuONP treatment. After treatment with CuONP of 100 μM, 60% of procaspase 3 and procaspase 9 decreased in SH-SY5Y, 40% reduced in H4 and 70% were cut down in PC12 cell lines. There were significant differences in expression in cells before and after treatment of CuONP.

### 3.6. CuONP Increased p53 Expression

To verify the induction of apoptosis by CuONP is through a p53-dependent pathway, we measured the p53 expression levels following treatment with CuONP on three different cell lines as described above. Figure 7 shows that CuONP upregulated p53 expression. After treatment with CuONP 100 μM, p53 expression increased. A 100% increase in p53 expression was observed in SH-SY5Y cells, a 200% increase in H4, and a 40% in PC12 cell lines. Each of the three cell lines responded differently to treatment with CuONP.

### 3.7. CuONP and L-BSO Reduced Thiol Level

To demonstrate the ROS involvement in CuONP neurotoxicity, we determined reduced thiols levels on the three cell lines (SH-SY5Y, H4 and PC12). It was determined that if pretreated with n-acetyl cysteine (NAC), an antioxidant [43,44], the damage can be prevented. Figure 8 shows the reduced thiol levels following treatment with CuONP or L-Buthionine sulfoximine (L-BSO), a potent inducer of experimental glutathione deficiency [45,46]. A.1, B.1, and C.1 shows that in the thiol level of CuONP treatment, the concentration of CuONP is 0.1 to 100 μM, and A.2, B.2, and C.2 shows the thiol level after treatment with L-BSO at 1 μM, and the concentration is 0.001 to 1 μM. The thiol level significantly decreased after cells were treated with CuONP and L-BSO at only at the highest dose, except the L-BSO started to work on H4 cells at 0.1 μM. There are significant changes to thiol levels but, if pretreated with NAC, the process of cell damage can be mitigated.

### 3.8. CuONP and L-BSO Induced Aβ Production

To find out the potential relationship between CuONP and AD pathology, we performed amyloid β-protein (Aβ) ELISA assay on SH-SY5Y and H4 cells. If cells were pretreated with NAC, the Aβ could not be detected. Figure 9 shows that the Aβ level after treatment with CuONP 100 μM or L-BSO 1 μM. CuONP increased Aβ40 and Aβ42, and L-BSO only increased Aβ42 in SH-SY5Y cells. CuONP and L-BSO can increase Aβ40, but not Aβ42 in H4 cells. When pretreated with NAC followed by the administration of the CuONP or L-BSO, there was no detectable increase in both SH-SY5Y and H4 cells. 

### 3.9. Differences between Cell Lines

Throughout the set of experiments performed in this study, there were notable differences between cell lines in terms of their responses to the various nanoparticles and other agents. Though this study did not examine the mechanistic differences between the different cell lines in terms of the observed responses, these differences are likely attributable to their different origins. PC12 is derived from rat adrenal pheochromocytoma that are notably nerve growth factor (NGF) receptive [47]. SH-SY5Y, originally called SK-N-SH, was derived from metastatic neuroblastoma tissue [48]. H4 cells were derived from human neuroglioma cells. Though the different cell lines are frequently used in neural studies, they are not identical, and some differences in results are to be expected.

## 4. Discussion

The defining characteristics of nanoparticles, their small size and large surface area to volume ratios, render nanoparticles more biologically active than larger materials. Due to their size, nanoparticles have a higher probability of reaching deep into the lungs, and once deposited, they seem to be able to translocate to other organs [49]. As the use of nanoparticles increases, it becomes more and more important to investigate their possible adverse effects to human health and the environment [50]. 

Metal oxide nanoparticles are one of the most important categories of nanoparticles. Some in vitro studies on the toxicity of metal oxide nanoparticles have been reported [51,52,53]. We compared three metal oxide nanoparticles: CuONP, ZnONP and Fe_2_O_3_NP cell cytotoxicity on three neuronal cell lines: SH-SY5Y, H4 and PC12. The MTS assay and Trypan Blue staining were used to measure cell cytotoxicity. We found that CuONP is highly cytotoxic to H4 and PC12 cells and moderately cytotoxic to Sy5y cells, while ZnONP and Fe_2_O_3_NP were not found to be cytotoxic to any of the three cell lines at the tested concentrations. Hanna et al. reported that CuONPs are highly toxic [14]. Other research has been performed on the toxicity of ZnONP, CuONP and TiO_2_NP on the bacteria *Vibrio fischeri* and the crustaceans *Daphnia magna* and *Thamnocephalus platyurus*. It was observed that the toxicity of TiO_2_NP is greatest, followed by CuONP and ZnONP [17]. Based on current research, it seems there is a high degree of variability in the response different cell types have to the toxic effects of nanoparticles [14].

Further, we consider the cytotoxicity of CuONP on the three cell lines. Apoptosis and necrosis are presently recognized as the two major types of physiologic or pathologic cell death. A prominent molecular hallmark of apoptosis is a specific pattern of internucleosomal fragmentation of DNA. Using a fluorescent marker, Terminal Deoxynucleotidyl Transferase (TdT), which is an enzyme that catalyzes the repetitive addition of dNTPs to the 3´-OH end of a DNA fragment, one can detect cells with apoptotic DNA fragmentation. We detected that only at the highest dose (100 µM), CuONP induced apoptosis in the three neuron cell lines, but not at lower doses (data not shown at lower doses). We used a TdT kit to stain SY5Y, H4 and PC12 cells after treatment with CuONP at 100µM and observed an increase in the number of apoptotic cells.

Caspase activation is a central feature of apoptosis. In general, caspase 8 and 9 appear to be “initiator” caspases, while caspase 3, 6 and 7 are secondarily activated and are termed “effector” caspases because they act on a variety of cell proteins. Procaspases are activated caspases through a process of self-cleavage. In this study it was found that procaspase 9 decreased after treatment of high dose CuONP, along with a corresponding reduction in procaspase 3. Further, we observed that levels of activated caspase 3 in cells increased dramatically. These results show that treatment with CuONP at 100 μM promotes apoptosis in cells. The decrease observed in SH-SY5Y cells is the smallest between the three cell lines, a result that is consistent with the cytotoxicity test results.

There are two general pathways for activation of apoptosis: the intrinsic and extrinsic pathways. The first is derived from extracellular signals and results in subplasmalemmal aggregation of caspases through an association of cytoplasmic domains of ligand receptors. The second is mediated by intracellular factors and involves changes in mitochondrial permeability, cytosolic release of cytochrome c, association of protein with an apoptosis-aggregating factor and activation of caspase 9 [54]. We also checked the expression of p53, as p53 triggers an extensive network of signaling pathways, to ensure that cells have an appropriate response to a given stress; basal levels of p53 are generally low in unstressed cells [55]. Intriguingly, p53 can intervene at every major step in apoptotic pathways: from extrinsic death receptor signaling, through the convergent pathway component Bid, to the intrinsic mitochondrial pathways involving apoptosome formation, and culminating in direct caspase activation [37,56]. Here, p53 expression increased after treatment of high dose of CuONP, indicating that p53 is involved in the apoptosis process induced by CuONP.

During the normal course of metabolism, oxygen is partially reduced as electrons leak out of the electron transport chain during respiration [57]. These partially reduced oxygen species can react with organic substances through non-catalytic means. Reactive oxygen species (ROS) are important cellular regulators but are also sources of considerable damage. Elevation of ROS beyond the buffering capacity of the cell can lead to oxidative stress. Elevated ROS levels can cause damage to DNA/RNA, proteins and lipids which may ultimately lead to apoptosis. Cells have developed several mechanisms to counteract elevated ROS levels such as a thiol reducing buffer, composed of components such as glutathione and thioredoxin, for the maintenance of the reduction-oxidation (redox) state of the cell, and enzymes to remove ROS (catalase, superoxide dismutase and glutathione peroxidase) [58]. Excess ROS production is also a notable component of some apoptosis pathways, with antioxidants exerting an inhibitory effect on the induction of cell death [59], meaning elevated ROS levels may be a sign of damage rather than the primary cause.

However, Previous research has shown that nanoparticles induce oxidative stress, which leads to inflammation [51,53]. In two studies about nanoparticles inducing cell apoptosis, it was found that TiO_2_ nanoparticles could not only photocatalyze DNA oxidative damage, but also induce cell apoptosis both in vitro and in human cells [60]. In Long et al., it was reported that nanoscale TiO_2_NP not only stimulated brain microglia to produce ROS through an oxidative burst, but the particles also interfered with mitochondrial energy production in vitro [61]. Based on the results of this study, it appears that CuONP has an effect of cell apoptosis on neuronal cells via ROS. L-BSO, which is a strong oxidant, had a similar effect on the cells to CuONP. Furthermore NAC, an antioxidant, appears to mitigate the damage from both L-BSO and CuONP.

The etiology and type of neuronal death in most neurodegenerative diseases remains controversial. However, research suggests that in neurodegenerative disorders such as AD, specific neuronal groups die by apoptosis [37]. As of now, it is clear that the AD etiology is related to numerous different factors. These include oxidative stress, neuroinflammation and other environmental factors. Based on prior research, nanoparticles have been found to induce brain inflammation [62]. This study has shown that CuONP induced apoptosis in three types of brain cells (SH-SY5Y, H4 and PC12), and CuONP neurotoxicity also involves ROS formation. These findings lend credence to the idea that CuONP exposure may be an environmental risk factor for AD.

AD is associated with extracellular deposition of amyloid Aβ in the form of senile plaques, intracellular accumulation of neurofibrillary tangles, and progressive neuronal loss [34]. Aβ40 and Aβ42, the latter of which is the most closely related to AD pathology, are produced from the cleavage of amyloid protein precursor (APP) by β and γ secretases. AD and the production of Aβ was linked by Anne Eckert to mitochondrial dysfunction, the caspase-apoptosis pathway and neuronal loss [36]. In this study, we observed increased Aβ production in neural cells undergoing apoptosis, which is consistent with the hypothesis. Neural cells exposed to CuONP experienced neurotoxicity and increased cell apoptosis, involving ROS and p53, and increased Aβ production. These results seem to suggest that exposure to CuONP may be an environmental factor involved in AD pathogenesis, and antioxidants such as NAC may mitigate its neurotoxic effect.

## 5. Conclusions

The experiments performed in this study affirm that CuONP exposure exerts neurotoxic effects on neuronal cells, while ZnONP and Fe_2_O_3_NP exhibited only limited toxic effects. SH-SY5Y, H4 and PC12 cells exposed to CuONP experienced significant increases in apoptotic events and significant decreases in cell proliferation, with H4 and PC12 experiencing the most profound effects. This is corroborated by corresponding increases in caspase 3 activity in all three cell lines, with H4 and PC12 cells again experiencing the most dramatic changes. Cells undergoing apoptosis experience a decrease in the activity of procaspases, as these procaspases are activated in apoptotic pathways; in all three cell lines, levels of procaspases 3 and 9 were found to be lower when exposed to CuONP. A final indicator of apoptosis, p53 expression, was also found to increase following CuONP treatments; this also shows that the apoptosis-pathway promoted by CuONP toxicity is likely p53-dependent. All of these results confirm that CuONP has the capacity to trigger apoptotic events in brain cells, though the magnitude of the response appears to be dependent on cell type. Additionally, exposure to CuONP decreased thiol levels in cells, an effect which was countered by pretreatment with the antioxidant NAC. This indicates CuONP toxicity induces ROS formation, but this effect was also highly dependent on cell type. Finally, Aβ40 and 42, The primary components of Aβ amyloid plaques that are pathological features of AD, were found to increase in SH-SY5Y cells when exposed to CuONP. This effect was also dampened with NAC pretreatment, indicating the Aβ increase may be due to oxidative stress. Notably, Aβ40 did not change in SH-SY5Y or H4 cells, and Aβ42 levels were mostly unchanged in H4 cells regardless of CuONP exposure or NAC pretreatment. Overall, these results support the idea that, via neuronal apoptosis, ROS generation and increases in Aβ levels, exposure to CuONP may be an environmental risk factor that contributes to AD etiopathology. Future in vivo study is necessary to confirm the potential link between CuONP exposure and AD. For public health concerns, regulation for environmental or occupational exposure of CuONP are thus warranted given AD has already become a pandemic.

## Figures and Tables

**Figure 1 ijerph-17-01005-f001:**
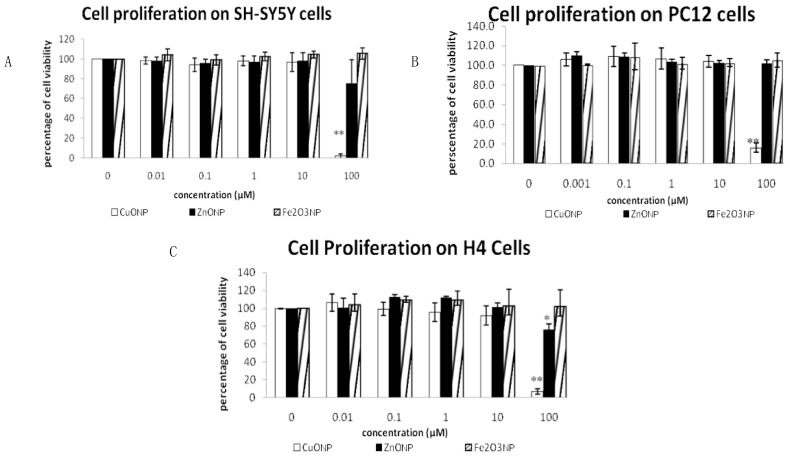
MTS assay. Effects of three nanoparticle in three cell lines: SH-SY5Y (**A**), H4 (**B**) and PC12 (**C**). Cells were plated in 96-well plate. The medium and fresh compound solutions were added after 24 h plating. After treated 48 h, cell viability was assessed by the MTS assay kit. Data are expressed as percentage of viable cells (mean ± SEM of three separate experiments, each experiment was performed in triplicate). * *p* < 0.05, ** *p* < 0.01.

**Figure 2 ijerph-17-01005-f002:**
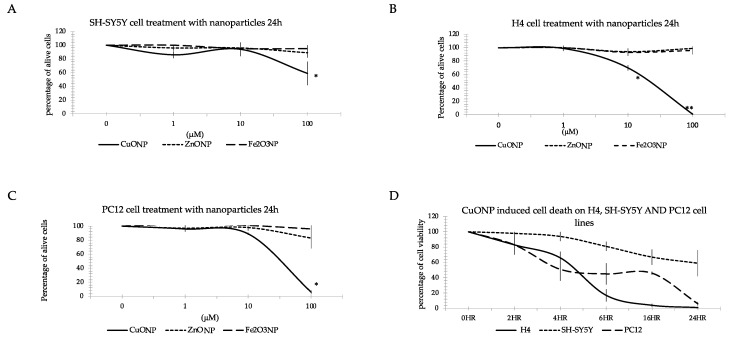
Trypan Blue staining. Effects of three nanoparticles in three cell lines: SH-SY5Y (**A**), H4 (**B**), and PC12 (**C**). Cells were plated in 24-well plate. The medium and fresh compound solutions were added after 24 h plating. After treated with the three NPs of CuONP, ZnONP and Fe_2_O_3_NP by the designed time, cells were harvested and resuspended it with medium and 0.4% Trypan Blue with a ratio of 1:2. We counted living cells by under microscope and compared with the control. Data are expressed as percentage of viable cells (mean ± SEM of three separate experiments, each performed in triplicate). * *p* < 0.05, ** *p* < 0.01.

**Figure 3 ijerph-17-01005-f003:**
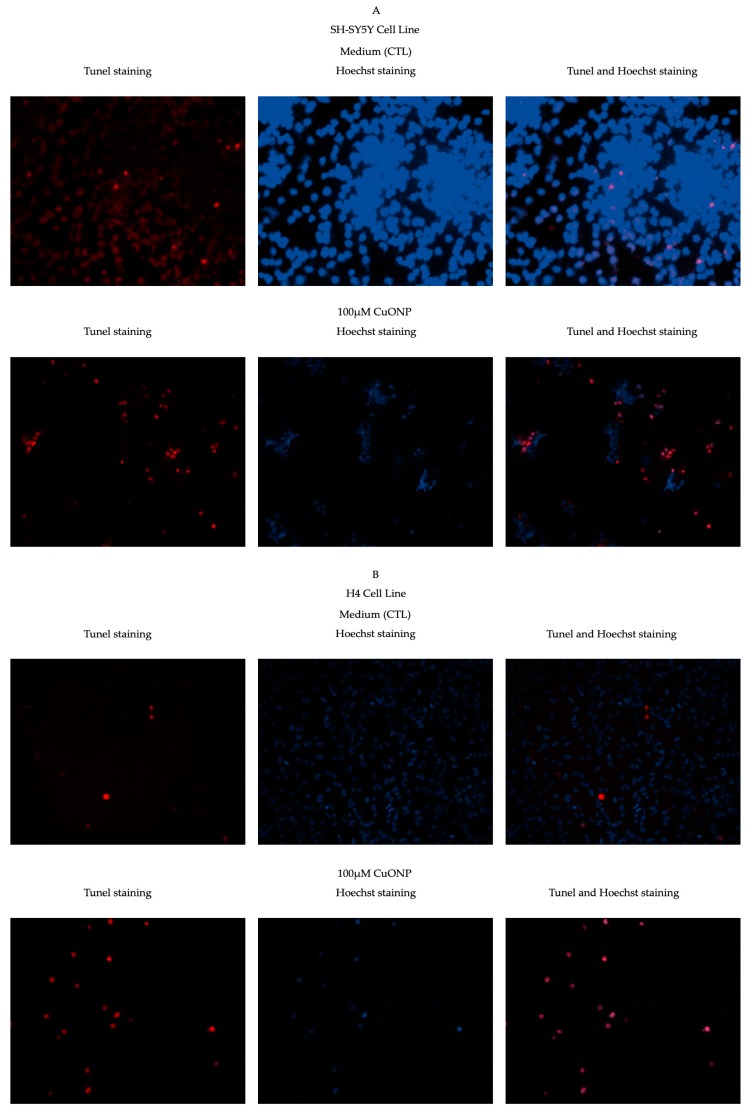
TdT-mediated dUTP nick-end labeling (TUNEL) staining. CuONP induced cell apoptosis in three cell lines: SH-SY5Y (**A**), H4 (**B**), and PC12 (**C**). Panel left: TUNEL staining, middle: Hoechst staining, and right: combination of TUNEL and Hoechst staining. Cells were plated in 8-chamber slides. The medium and fresh compound solutions were added after 24 h plating. After treatment with CuONP for 24 h, cells were fixed, permeabilized, and then incubated with terminal deoxynuceotydyl transferase. For total cell counting, cells were stained by Hoechst. Pictures were taken with a fluorescent microscope and numbers of TUNEL-positive cells were counted; (**D**) percentage of TUNEL positive cells in total cells in the three cell lines. ** *p* < 0.01.

**Figure 4 ijerph-17-01005-f004:**
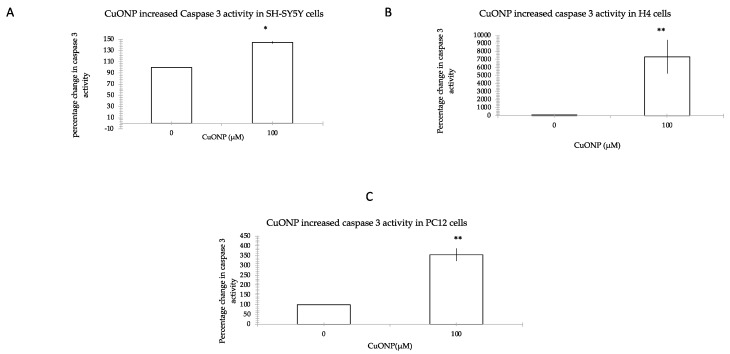
Caspase 3 activity assay. CuONP induced cell apoptosis by activating caspase 3 in three cell lines: SH-SY5Y (**A**), H4 (**B**), and PC12 (**C**). After treatment with CuONP, cells were harvested by assay buffer. Then, 2.0 × 10^6^ cells were lysed using assay. The caspase 3 fluorometric substrate (Ac-Asp-Glu-Val-Asp-AMC) was added to the lysate. We measured the fluorescence of the cleavage product over time at room temperature for 1 h and 30 min. Maximal enzyme activity (*V*_max_) was calculated and expressed as relative fluorescence units/second. Comparison with the control, percentage of increasing caspase 3 activity was calculated. * *p* < 0.05, ** *p* < 0.01.

**Figure 5 ijerph-17-01005-f005:**
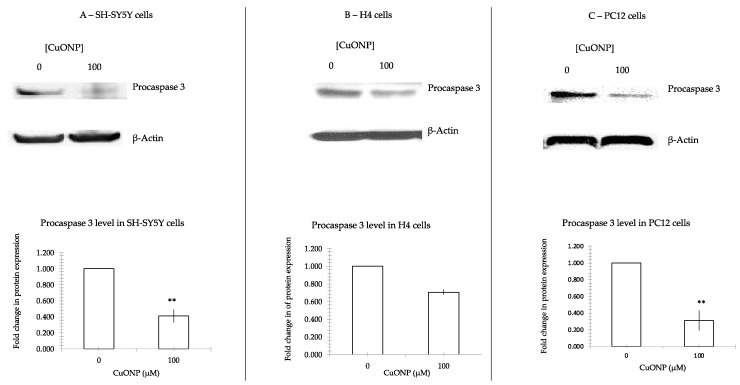
Procaspase 3 protein expression. CuONP activates cell apoptosis pathway by auto-cleaving procaspase 3 into active caspase 3. We incubated CuONP of 100 μM with three lines of cells: SH-SY5Y (**A**), H4 (**B**), and PC12 (**C**). Cells were plated in 60 mm dish and treated with CuONP for 6 h in H4 and PC12 cells, and 16 h in SH-SY5Y cells. Cells were subjected to lyses in a RIPA buffer. The same amount of protein was run in 4%–12% Tris-Bis gel electro-transferred to nitrocellulose membrane. After being washed with PBST, the membranes were incubated with the first antibody 1:200 overnight and the secondary antibody 1:5000 for 2 h and developed a film by detection kit. The membranes were stripped by stripped buffer and re-incubated with first antibody actin to control the protein amounts. The films were scanned, and densities were analyzed by NIH ImageJ software. ** *p* < 0.01.

**Figure 6 ijerph-17-01005-f006:**
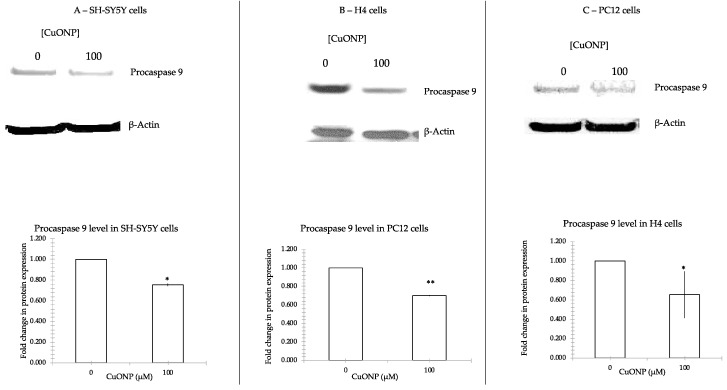
Procaspase 9 protein expression. CuONP activates cell apoptosis pathway by auto-cleaving procaspase 9 into active caspase 9. We incubated CuONP 100 μM with three cell lines: SH-SY5Y (**A**), H4 (**B**), and PC12 (**C**). The same amount of protein was run in 4%–12% Tris-Bis gel and electro-transferred. The membranes were incubated with the first antibody 1:200 overnight and the secondary antibody 1:5000 for 2 h and developed a film by detection kit. The membranes were stripped by stripped buffer and re-incubated with the first antibody β-Actin as a control for the protein loading. The films were scanned and densities were analyzed by NIH ImageJ software (NIH, Bethesda, MD, USA). * *p* < 0.05, ** *p* < 0.01.

**Figure 7 ijerph-17-01005-f007:**
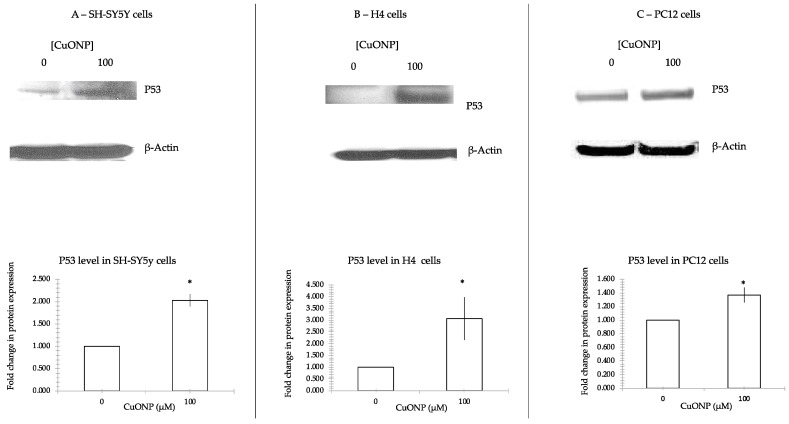
p53 protein expression. CuONP activated p53 and induced cell apoptosis. We incubated CuONP at 100μM with three cell lines: SH-SY5Y (**A**), H4 (**B**), and PC12 (**C**). Cells were plated in 60 mm dish and treated with CuONP for 6 h in H4 and PC12 cells, and 16 h in SH-SY5Y cells. Cells were subjected to lyses in a RIPA buffer. The same amount of protein was run in 4%–12% Tris-Bis gel electro-transferred to nitrocellulose membrane. After washing with PBST, the membranes were incubated with the first antibody p53 1:1000 overnight and the secondary antibody 1:5000 for 2 h and developed a film by detection kit. The membranes were stripped by stripping buffer and re-incubated with the first antibody β-Actin as control for the protein loading. After incubation with enhanced chemiluminescence (ECL) solution, membranes were imaged and densities were analyzed by NIH ImageJ software. * *p* < 0.05.

**Figure 8 ijerph-17-01005-f008:**
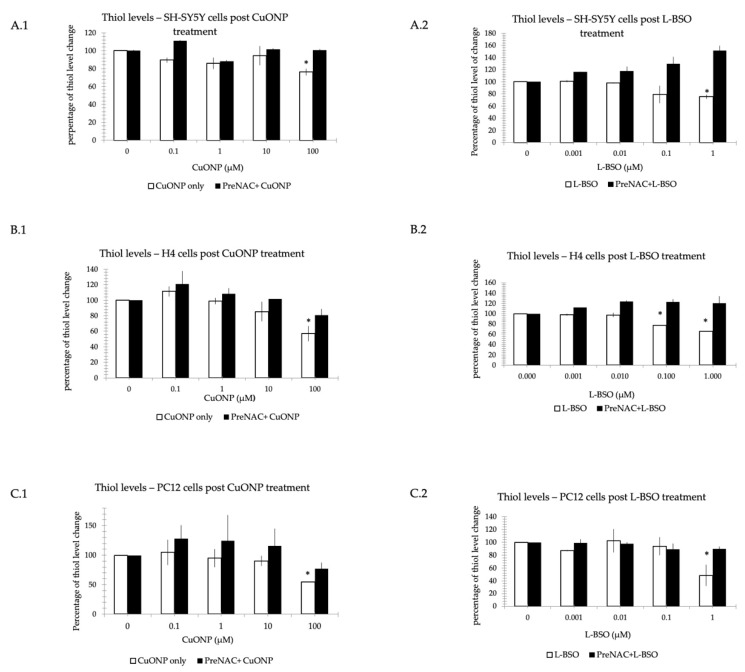
Thiol level reduction assay. Nanoparticles reduced thiol level. We incubated CuONP (CuONP: 0.1 to 100 μM or L-BSO: 0.001 to 1 μM) with three cell lines: SH-SY5Y (**A**), H4 (**B**), and PC12 (**C**): A.1, B.1, C.1 cells were treated by CuONP, and L-BSO was incubated with cells showed on A.2, B.2, C.2. Cells were treated by n-acetyl cysteine (NAC) 5 mM for 1 h, then were added into CuO or L-BSO of different concentrations separately. Cells were plated into 6-well plate and treated with CuONP or L-BSO: 6 h on H4 and PC12 cells, and 16 h on SH-SY5Y cells. Then, 2 × 10^5^ was harvested and washed with PBS twice. Cells were subjected to lyses in 200 μL lysis buffer. Mixture dye and cell lysates were added into 96-well black plate. The plate was read immediately with excitation at 485 nm and emission at 538 nm. The percentage of fluorescent changes were calculated. * *p* < 0.05.

**Figure 9 ijerph-17-01005-f009:**
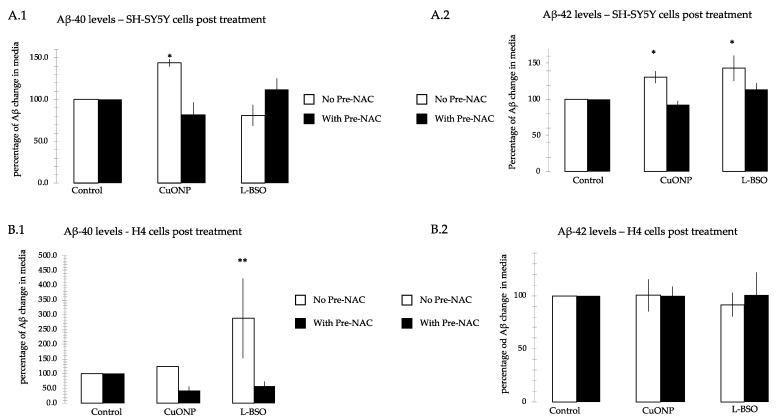
Aβ40 and Aβ42 ELISA. We measured Aβ40 and Aβ42 levels in cell media after treatment with CuONP at 100 μM and L-BSO at 1 μM, and they were also measured when cells were pretreated with NAC for 1 h, followed by treatment of CuONP and L-BSO, respectively. Measurements were taken of SY5Y(A) levels for Aβ40 (**A.1**) and Aβ42 (**A.2**) and of H4 levels for Aβ40 (**B.1**) and Aβ42 (**B.2**). Aβ40 and Aβ42 levels were increased by treatment with CuONP, and only Aβ42 was increased by treatment with L-BSO on SH-SY5Y cells; Aβ40 level was increased by treatment with CuONP and L-BSO, but not Aβ42. Cells were plated into 6-well plate and treated with CuONP or L-BSO in 1.5 mL media: 6 h on H4 cells, and 16 h on SH-SY5Y cells. Media were harvested and spun down. The supernatants were saved for ELISA assay. For assay, samples were mixed with antibody according to the protocol and incubated overnight. Substrates were added into the plate and read with luminescent reader. The percentage of fluorescent changes were calculated. * *p* < 0.05, ** *p* < 0.01.

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
