# Peer review of "Exposure of CuO Nanoparticles Contributes to Cellular Apoptosis, Redox Stress, and Alzheimer’s Aβ Amyloidosis"

_ijerph, 2020, doi:10.3390/ijerph17031005_

Round 1
Reviewer 1 Report
The manuscript by Shi and colleagues presents novel data indicating toxicity of CuO towards 3 neuronal cell lines. The data is convincing and the paper well written, although there are some grmmatical issues that need attention. Experiments in animal models would support the conclusions drawn in the paper, and greatly elevate the importance of the results to in vivo neurodegeneration.
Title: Suggests animal or human study, while data presented is in vitro.
Abstract: Include summary of amyloid results in abstract as indicated in the title of the paper.
Introduction:
"There is also an opportunity for the inhalation of them by workers during their manufacture [2, 10]." Change to: There is also an opportunity for their inhalation by workers during their manufacture [2, 10].
"Thus, it appears that toxicity to the neuro cells were greater than toward bacteria, while the opposite was true for T-cells." Include bacteria toxicity level in paragraph for comparison.
Line 108: "...in triplicate."
Results:
What is the concentration of CuO in Figure 2D? Y axis is missing?
Figure 4B: Decrease Y-axis numbering (and for many other figures).
Grammar: After treatment with CuONP, 20% SH-SY5Y and almost 60% H4 and PC12 cells displayed TUNEL-positive staining, whereas less than 1% of the control cells were TUNEL-positive. CuONP induced cell apoptosis of all three cell lines after 24 h treatment at 100 μM.
Figure 3D: Improve low resolution figure.
Figure legend 5 and 6: Activate or active?
Formatting of figures throughout requires much attention.
Discussion:
How likely is it for neurons to encounter CoO at concentrations of 100 micromolar or greater?
Grammar needs a little help throughout.
Thiol level reduction might be expected in dying cells independent of ROS production.
Mechanistic cellular level conclusions are correct. However, temper CuO conclusions re Alzheimer's disease in Discussion. Only in vitro results are presented, and that is a long way away from an animal model or human scenario.
Author Response
Responses to Reviewer 1
The manuscript by Shi and colleagues presents novel data indicating toxicity of CuO towards 3 neuronal cell lines. The data is convincing and the paper well written, although there are some grmmatical issues that need attention. Experiments in animal models would support the conclusions drawn in the paper, and greatly elevate the importance of the results to in vivoneurodegeneration.
“Title: Suggests animal or human study, while data presented is in vitro.”
Response: We have altered the title to more accurately reflect the work.
Abstract: Include summary of amyloid results in abstract as indicated in the title of the paper.
Response: We have included Aβ level results.
“Introduction:
"There is also an opportunity for the inhalation of them by workers during their manufacture [2, 10]." Change to: There is also an opportunity for their inhalation by workers during their manufacture [2, 10].
"Thus, it appears that toxicity to the neuro cells were greater than toward bacteria, while the opposite was true for T-cells." Include bacteria toxicity level in paragraph for comparison.
Line 108: "...in triplicate."”
Response: We have made the requested changes, thank you for pointing them out.
“Results:
What is the concentration of CuO in Figure 2D? Y axis is missing?
Figure 4B: Decrease Y-axis numbering (and for many other figures).
Grammar: After treatment with CuONP, 20% SH-SY5Y and almost 60% H4 and PC12 cells displayed TUNEL-positive staining, whereas less than 1% of the control cells were TUNEL-positive. CuONP induced cell apoptosis of all three cell lines after 24 h treatment at 100 μM.
Figure 3D: Improve low resolution figure.
Figure legend 5 and 6: Activate or active?
Formatting of figures throughout requires much attention.”
Response: Figures, fonts and legends have been edited based on the reviewer’s critiques and suggestions.
“Discussion:
How likely is it for neurons to encounter CoO at concentrations of 100 micromolar or greater?”
Response: It is fairly unlikely for neurons to encounter CuO, or Cu2+ at that concentration in vitro. We will adjust the concenntrations in the future in vivo studies.
“Grammar needs a little help throughout.
Response: We have edited the whole text to remove the grammatic errors.
Thiol level reduction might be expected in dying cells independent of ROS production.”
Response: We have not found anything to suggest this the case, and the relationships between ROS and thiol levels, as well as CuONPs effect on ROS generation, are fairly well established. It is true that elevation of ROS is involved in various apoptotic pathways, bringing up the possibility that the observed elevation of ROS following CuONP exposure may have been a signal/mediator, rather than the cause of damage. we have noted this aspect in the discussion of the text, thank you for pointing this out.
“Mechanistic cellular level conclusions are correct. However, temper CuO conclusions re Alzheimer's disease in Discussion. Only in vitro results are presented, and that is a long way away from an animal model or human scenario.”
Response: We have altered some of the wording in the discussion directly relating our results to Alzheimer’s disease. Future in vivo studies to verify our in vitro results are under way.
Reviewer 2 Report
In this paper Ying Shi et al. investigated the effect of exposure of copper oxide nanoparticles on three different neuronal cell lines.
The work is well done and well written. However, I would suggest several major changes before its publication.
The title is a little bit ambitious, on my opinion. The experiments reported shown a plethora of effects of NPs on cells. I would include the formation of A-beta among the toxic effects. Introduction: the motivation of the choice of the three cell lines is missing. A) Please specify what kind of thiol detection kit has been employed. B) L-BSO has to be explained in 2.7 paragraph and a reference has to be cited. Figure 1.Please change the pattern of Fe2O3NP bars. The author may suggest an explanation for the different behaviors of the three cell lines? The results obtained on Cu NPs are not surprising because of the role of Cu(I) and Cu (I) in cells (Cu(II) can be easily reduced to Cu(I) in the reducing environment of the cell). This part should be discussed as added value of this work (please see the role of copper in V. Angeli et al./Archives of Biochemistry and Biophysics 481 (2009) 191–196 and Archives of Biochemistry and Biophysics Volume 487, Issue 2, 15 July 2009, Pages 146-152). Line 300: please give references for the employment of NAC as antioxidant and L-BSO as inducer of GSH deficiency. Figure 8 is complex. I would suggest to make the effort of producing a kind of “summary figure” resuming all the results. Figure 9. Based on the differences among H4 and SH-SY5Y cell lines, please try to discuss and give referenced explanation of these differences. Line 387: give e reference. Lines 388- 393: this part is redundant and well known. Lines 418-424:this part has many repetitions. Please, try to summarize.
Several typing errors have to be corrected (bold lines 106-108; fluoresce line 130; line 350).
Author Response
Responses to Reviewer 2
Thanks for your comments and time.
In this paper Ying Shi et al. investigated the effect of exposure of copper oxide nanoparticles on three different neuronal cell lines.
The work is well done and well written. However, I would suggest several major changes before its publication.
“The title is a little bit ambitious, on my opinion.”
Response: We have altered the title to more accurately reflect our work.
“The experiments reported shown a plethora of effects of NPs on cells. I would include the formation of A-beta among the toxic effects.”
Response: We have included a mention of Aβ level increase.
“Introduction: the motivation of the choice of the three cell lines is missing.”
Response: We have added a brief section in the introduction regarding the choice of cell lines in our studies.
“A) Please specify what kind of thiol detection kit has been employed.”
Response: The name and manufacturer of the thiol detection kit is listed in section 2.7 of the Materials and Methods section; it was the “Fluorescent Thiol Detection Kit” manufactured by Cell Technology Inc.
“B) L-BSO has to be explained in 2.7 paragraph and a reference has to be cited.”
Response: We have explained L-BSO briefly in the Materials and Methods section, and have cited references for its GSH-depletion effect.
“Figure 1.Please change the pattern of Fe2O3NP bars.”
Response: We have kept the original pattern in the figure. However, we have enlarged each graph of the figure, and it should be more clear than before.
“ The author may suggest an explanation for the different behaviors of the three cell lines?”
Response: We did not go into the specific mechanistic differences in this study. However, a section was added highlighting the different origins of the three cell lines.
“ The results obtained on Cu NPs are not surprising because of the role of Cu(I) and Cu (I) in cells (Cu(II) can be easily reduced to Cu(I) in the reducing environment of the cell). This part should be discussed as added value of this work (please see the role of copper in V. Angeli et al./Archives of Biochemistry and Biophysics 481 (2009) 191–196 and Archives of Biochemistry and Biophysics Volume 487, Issue 2, 15 July 2009, Pages 146-152).”
Response: We have added a brief explanation of the role of copper ions in cellular redox reactions. Thank you for the provided reference.
“Line 300: please give references for the employment of NAC as antioxidant and L-BSO as inducer of GSH deficiency.”
Response: We have added the requested references.
“Figure 8 is complex. I would suggest to make the effort of producing a kind of “summary figure” resuming all the results.”
Response: Figure 8 had some inconsistent legend entries that made it seem more complicated than it is. This has been rectified in the current version.
“Figure 9. Based on the differences among H4 and SH-SY5Y cell lines, please try to discuss and give referenced explanation of these differences.”
“Line 387: give e reference.”
Response: The requested reference has been added.
“Lines 388- 393: this part is redundant and well known.”
Response: We have summarized/condensed this particular section.
“Lines 418-424:this part has many repetitions. Please, try to summarize.”
Response: We have reworded this section.
“Several typing errors have to be corrected (bold lines 106-108; fluoresce line 130; line 350).”
Response: We have corrected these errors.

Reviewer 3 Report
Overall content of the manuscript provides an in depth look at the role of copper oxide nanoparticles in AD. The author found that CuONP is highly cytotoxic to H4 and PC12 cells and moderate cytotoxic to Sy5y cells, while ZnONP and Fe2O3NP were not found to be cytotoxic to any of the three cell lines at the tested concentrations. However, a few more areas need to be addressed before publication.
What is the reason for the higher toxicity of CuONP than ZnONP and Fe2O3NP? Why CuONP is highly cytotoxic to H4 than other cell lines? Line 28, define ASTM. Give details of the statistical methods used. legend 2c not matched with fig. What is the difference between fig 2b and 2c? No Y-axis (Fig 2D). Give a clear description of fig 2. Which image software is used? Figure 6, Y-axis is missing. 100 µM of nanoparticle exposure may also induce necrosis. Does the author check the narcosis marker?Author Response
Responses to Reviewer 3
Overall content of the manuscript provides an in depth look at the role of copper oxide nanoparticles in AD. The author found that CuONP is highly cytotoxic to H4 and PC12 cells and moderate cytotoxic to Sy5y cells, while ZnONP and Fe2O3NP were not found to be cytotoxic to any of the three cell lines at the tested concentrations. However, a few more areas need to be addressed before publication.
“What is the reason for the higher toxicity of CuONP than ZnONP and Fe2O3NP? Why CuONP is highly cytotoxic to H4 than other cell lines?”
Response: We have added an explanation for the heightened toxicity of CuONP. We did not address the mechanisms behind the differences between cell lines in this study as it is beyond the scope of current studies.
“Line 28, define ASTM.”
Response: ASTM is the current full name of what was formerly known as the American Society for Testing and Materials; we have made a note of this in the text, but it is no longer an acronym.
“Give details of the statistical methods used.”
Response: Much of the work was done using Excel (which has been clarified) and other calculated values are described in text.
“legend 2c not matched with fig. What is the difference between fig 2b and 2c? No Y-axis (Fig 2D). Give a clear description of fig 2.”
Response: Figure 2 had a repeated section (the H4 cell line graph was duplicated, and the PC12 cell line graph was missing), which likely contributed to this confusion. We have rectified this, and the figure should now be more clear. Thank you for pointing this out.
“Which image software is used?”
Response: The final films were scanned, and densities were analyzed by NIH ImageJ software. We indicated this in the Western Blotting procedure and relevant figure captions.
“Figure 6, Y-axis is missing.”
Response: This has been rectified, thank you for pointing this out.
“100 µM of nanoparticle exposure may also induce necrosis. Does the author check the narcosis marker?”
Response: We did not check the necrosis marker in this set of experiments, but we will check this in our future studies.

Round 2
Reviewer 2 Report
The paper has been modified according to my revision. The paper is ACCEPTABLE for publication.